



**Ionospheric Pc1 waves during a storm recovery phase**
**observed by CSES**
Xiaochen Gou[1], Lei Li[1*], Yiteng Zhang[1*], Bin Zhou[1], Yongyong Feng[1], Bingjun Cheng[1]
Ji Liu[1], ZhiMa Zeren[2], Xuhui Shen[2]
[1]State Key Laboratory of Space Weather, National Space Science Center, Chinese
Academy of Sciences, Beijing 100190, China;
[2]Institute of Crustal Dynamic, China Earthquake Administration, Beijing 100029, China;
**Abstract**
During the storm recovery phase on August 27, 2018, the China Seismo-
Electromagnetic Satellite (CSES) detected Pc1 wave activities both in the Northern and
Southern hemispheres in the high latitude post-midnight ionosphere with a central
frequency about 2 Hz. Meanwhile, the typical Pc1 waves were simultaneously
observed by the Sodankylä Geophysical Observatory (SGO) stations on the ground for
several hours. In this paper, we study the propagation characteristics and possible
source regions of those waves. Firstly, we find that the satellites observed Pc1 waves
exhibit mixed polarization and the wave normal is almost parallel with the background
magnetic field. The field-aligned Poynting fluxes point downward in both hemispheres,
implying the satellites are close to the wave injection regions in the ionosphere at
about $L$=3. Furthermore, we also find that the estimated position of the plasmapause
calculated by models is almost at $L$=3. Therefore, we suggest the possible sources of
waves are near the plasmapause, which is consistent with previous studies that the
outward expansion of the plasmasphere into the ring current during the recovery
phase of geomagnetic storms may generate electromagnetic ion cyclotron (EMIC)
waves and then these EMIC waves propagate along the background magnetic field
northward and southward to the ionosphere at about $L$=3. Additionally, the ground
station data show that Pc1 wave power attenuates with increasing distance from $L$=3,
supporting the idea that CSES observes the wave activities near the injection region.
The observations are unique in that the Pc1 waves are observed in the ionosphere in
nearly conjugate regions, where transvers Alfven waves propagate down into the
ionosphere.

**1 Introduction**
Electromagnetic ion cyclotron (EMIC) waves are in the typical frequency range of 0.1–
5Hz which corresponds to Pc1 pulsations on the ground. Generally, in the





magnetosphere, EMIC wave can be excited by cyclotron instability of hot ions (1-100
keV) with temperature anisotropy ($T_\perp > T_{//}$) near the Earth's magnetic equator,
particularly, in the region with large plasma density and weak magnetic field, such as
the plasmapause, ring current and plasma sheet [Cornwall et al., 1965; Erlandson et
al.,1993; Horne and Thorne, 1993; Anderson et al., 1996; Lin et al., 2014]. Previous
studies indicate that hot ion temperature anisotropy ($T_\perp > T_{//}$) near the Earth's magnetic
equator can be caused by several possible mechanisms, such as plasmapause
expanding into ring current region during storm recovery phase [Wentworth,1964;
Russell & Thorne, 1970], mid-energy ions penetrating into the ring current region from
the plasma sheet [Bossen et al., 1976], the solar wind dynamic pressure enhancement
or the magnetosphere compression [Olson & Lee, 1983; Anderson & Hamilton, 1993;
McCollough et al., 2010; Usanova et al. 2012]. Statistical results show that EMIC waves
are associated with magnetic activities and have a peak occurrence during the storm
recovery phase [Wentworth, 1964; Erlandson & Ukhorskiy, 2001; Bortnik et al., 2008].

Generally, EMIC waves are excited at or near the Earth's magnetic equator, and then
propagate along the background magnetic field toward the high latitude region, can
penetrate into the upper ionosphere under certain conditions. The left-hand polarized
(LHP) Alfvén waves incident from the magnetosphere can couple to the right-hand
polarized (RHP) compressional, isotropic waves in the ionosphere by the anisotropic
ionospheric Hall currents [Fraser et al., 1975a, 1975b; Fujita and Tamao 1988]. Since
the wavelength of EMIC waves with frequency about 1Hz is comparable with the scale
size of the ionospheric minimum in the Alfven speed, they can be trapped and ducted
in this region of low Alfven speed [Lysak et al., 1999]. Thus, the EMIC waves can be
observed both at the low earth orbit (LEO) and on the ground as Pc1 geomagnetic
pulsations with different characteristics.

At ionospheric altitudes, satellite observations of Pc1 waves are usually provided by
the onboard magnetometers. MAGAT observed Pc1 waves at an ionospheric altitude
of 350-550km, with both LH and RH polarizations in a latitudinally narrow (<100 km)
region [Iyemori and Hayashi, 1989]. In recent years, with the development of LEO
satellites, various statistical studies of EMIC waves have been carried out to reveal the
global propagation characteristics, spatial distribution, and geomagnetic dependence
of Pc1 waves. According to the statistical analysis of CHAMP satellite data during one
solar cycle, Park et al. [2013] found that Pc1 waves are mostly linearly polarized, having
a peak occurrence at subauroral latitudes, and weakly dependence on the magnetic
activity and the solar wind velocity. The SWARM data show a peak occurrence rate of
Pc1 waves at middle latitude including sub-auroral zone. Moreover, these waves are


linear polarization dominated, propagating oblique to the background magnetic field,
preferable to occur during late recovery phase of the storm [Kim et al. 2018a].
In this paper, we report a Pc1 wave event observed by the China Seismo-
Electromagnetic Satellite (CSES), as well as the SWARM satellite. Based on both electric
and magnetic field measurements, we study the propagation characteristics and
possible source regions of those Pc1 waves occurring at high latitude in the Northern
and Southern hemisphere ionosphere during the recovery phase of the geomagnetic
storm on 25-28 August 2018.

**2 Data sources**
The China Seismo-Electromagnetic Satellite (CSES) was launched on February 2, 2018,
into a sun-synchronous circular orbit at an altitude of 507 km with an inclination angle
of 97.4°. The local time of the descending node is 14:00. We use the magnetic field
data from the High Precision Magnetometer (HPM) and the electric field data from the
Electric Field Detector (EFD) onboard CSES. HPM includes two three-components
fluxgate sensors to collect vector magnetic field data with a sampling rate of 60Hz, and
the noise of the sensors are less than 0.02nT /√Hz @1 Hz [Zhou et al., 2018; 2019].
EFD consists of four spherical sensors, which can realize three-components electric
field detection at a broad frequency range from DC to 3.5MHz, in which the ULF band
provides 125Hz sampled waveform signal [Huang et al., 2018]. SWARM was launched
on November 22, 2013, which has three satellites at altitudes of 450 − 550 km with an
inclination angle of 88°. In addition, we also use the geomagnetic data from Sodankylä
Geophysical Observatory (SGO) stations, the solar wind data of OMNI from CDA Web
and Dst index from WDC Web.

**3 Observations**
Figure 1 shows the variation of solar wind parameters and the geomagnetic index
during the Pc 1 wave event in this study. The Dst index, interplanetary magnetic field,
solar wind speed and solar wind dynamic pressure from Aug. 25 to 29, 2018 are shown
from top to bottom. It can be seen that during the magnetic storm, the Dst index
decreased to -170 nT at 8:00 26 August. On the days from 27 to 28 August, the
interplanetary magnetic field (IMF) was northward, Dst experienced a minor increase.
The Pc1 waves were observed by CSES and SWARM between UTC 22:50 − 23:30
(marked by the black line in Figure 1) in the magnetic storm recovery phase on Aug.
110  27, 2018.



## 3.1 Spatial-temporal characteristics of Pc1 waves

On Aug. 27, 2018, CSES and SWARM-A satellites passed through the ionospheric Pc1 wave regions for three times, in the Northern and Southern hemispheres, marked by squares (CSES) and triangles (Swarm) in Figure 2. Firstly, at around UTC 23:00 (local time about 02:06 to 02:34), SWARM-A and CSES satellites were located at geomagnetic latitude about 56° S~53° S, at $L$ shell region about 3.0 ~ 3.4, with a distance about 300km apart, both successively observed Pc1 waves in the Southern hemisphere, as shown in Figure 2. SWARM-A observed the Pc1 waves at about UTC 22:50 (QD-LAT=56° S, $L$=3.4) about 10 minutes before CSES, with a maximum amplitude about 12 nT and a central frequency about 2 Hz, lasting for 1 minute, as shown in Figure 3. Then, CSES observed the Pc1 wave at UTC 23:02 (QD-LAT=54°S, L=3.1) by the magnetometer HPM (shown in Figure 4), with a maximum amplitude about 1.5 nT and a central frequency about 2 Hz, lasting a minute and a half. Thereafter, at about UTC 23:30 (local time about 01:27 to 01:22), the CSES flew away to the Northern hemisphere, passing through the Pc1 wave region again at geomagnetic latitudes about 54° N, $L$ values about 3.1. As shown in Figure 5, the maximum amplitude is about 10 nT and the central frequency is about 2 Hz, with a duration about 1 minute. Around this time, since the SWARM satellite was about 6000 km northeast of the CSES satellite, no Pc1 waves were observed by SWARM.

At the same time, the typical Pc1 waves were also observed by the SGO stations on the ground for several hours. As shown in Figure 6, from UTC 22:00 to 24:00, SGO stations recorded continuous pulsations with a central frequency of about 2-3 Hz. In Figure 6, from top to bottom are the observations from SGO stations: Sodankylä (SOD; $L$ = 5.3, 64.3°N, 105.6°E, QD), Oulu (OUL; $L$ = 4.5, 61.9°N, 104.1°E, QD), and Nurmijärvi (NUR; $L$ = 3.4, 57.1°N, 101.2°E, QD) from ~21:00 to 24:00 UT. The wave power of Pc1 pulsations increases monotonically with the decrease of $L$ shell values of SGO stations, with the maximum power at NUR station, which is close to the region where CSES observed Pc1 in the Northern hemisphere. Because of the ducting effect of Pc1 waves in the ionospheric waveguide, Pc1 waves are likely to be seen at a long distance away from the source region [e.g., Fujita and Taomao, 1988; Kim et al., 2010]. Since the boundary of the waveguide is not a perfect conductor, some absorption may happen when waves propagate in the waveguide, resulting in attenuation of the wave power. So, comparing the wave power observed by different ground stations, it is possible to infer the probable location of the wave source. Therefore, in our case, we suggest that the injection source region of the Pc1 waves in the Northern hemisphere should be near (QD-LAT=54 - 56°N, $L$=~ 3.3), where CSES and NUR observed the pulsations, and

after incidence on the ionosphere, the waves were ducted toward northeast, observed
by the ground stations located at higher latitudes.

**3.2 Propagation characteristics of Pc1 waves**
Wave polarization is another property that provides information on the wave source
and spatial characteristics of wave propagation. According to theoretical studies, the
incident LPH Alfven waves in the ionosphere can gradually change to RHP as the waves
propagate in the ionosphere away from the injection region [e.g., Fujita and Taomao
1986]. Close to the injection region, the polarization pattern is usually complex,
because the waves near the injection source are combined with incident waves and
ducting waves [Hayashi et al., 1981; Kim et al., 2010].

We further analyzed the propagation characteristics of Pc1 waves observed by CSES
and SWARM satellites in the Northern and Southern hemispheres during the magnetic
storm recovery phase. Firstly, we converted the magnetic field into field-aligned
coordinates (FAC) and then applied polarization analysis according to the method of
Means et al. [1972]. Figure 7, from top to bottom, shows SWARM magnetic field
components in FAC (including perpendicular components Br and Ba marked in blue
and green, parallel component Bz marked in red), magnetic wave power spectrum in
perpendicular direction and parallel direction, wave normal angle (0° indicates parallel
propagation and 90° indicates perpendicular propagation to the background magnetic
field), ellipticity (positive indicates RHP and negative indicates LHP). For CSES, electric
components in FAC, electric wave power spectrum in perpendicular direction and
parallel direction, and field-aligned Poynting flux are also included in Figure 8-9.

It can be seen from the SWARM and CSES data in the Southern (Figure 7,8) and
Northern hemispheres (Figure 9), the wave normal angles predominate below ~ 20°,
indicating that Pc1 waves almost parallel propagate with the background magnetic
field. Our result is somewhat different from the nightside observations in the
ionosphere by Pisa et al. (2015) and Kim et al. (2018), which show the wave normal
angles are scattered or have different tendency between two hemispheres. For CSES,
based on the HPM and EFD data, we also calculate the field-aligned Poynting flux of
Pc1 waves (shown by the blue lines in the bottom panels of Figure 8 and 9), which is
positive in the Northern hemisphere, negative in the Southern hemisphere, indicating
that Pc1 waves observed by CSES propagate along the background magnetic field
downward into the ionosphere in the both hemispheres.

On the other hand, we find that the waves have dominant perpendicular power, and





the parallel power (compressional power) is almost zero, which means the waves are
transverse. The transverse wave is one of the characteristics of the incident wave near
the wave injection region [Engebretson et al.,2008; Kim et al., 2010]. The transverse
wave also explains why the downward component in the local North-East-Down
coordinates has the minimum wave power, as observed by satellites and ground
stations (Figure 2-3, Figure 6). Near the injection region with a geomagnetic latitude
of ~55°, the dip angle of the geomagnetic field is about 73°. For a transverse wave, the
power projected to the downward direction should be small. We further find the wave
normal, electric field vector, background magnetic field are almost lie in the same
plane (not shown here) with a deviation less than +/- 8°, which proves that the incident
transverse wave is Alfvénic.
And the ellipticity of Pc1 waves shows mixed polarization for the waves detected by
CSES and SWARM in both hemispheres. To check whether our calculation results truly
represent these wave properties, we also use Minimum and Maximum Variance
Analysis (MVA) to get the MVA hodograph and the wave normal direction (not shown
here), which are also consistent with current results. Therefore, it seems that all the
Pc1 waves observed by CSES and SWARM have mixed ellipticities and propagate along
the background magnetic field.

**Discussion**
In 1964, Wentworth et al. proposed that during storm recovery phase, the
plasmapause expanding into the ring current region can excite EMIC wave. Through
simulation, Horne and Thorne et al. [1993] found that the growth rate of EMIC wave
inside the plasmapause is obviously lower than that outside the plasmapause, and its
peak is near the plasmapause. Using the CLUSTER satellite data, Grison et al. [2013]
observed a typical magnetospheric EMIC wave near the plasmapause at an *L* value of
about 4.2. They also believe that the Pc1 wave is easy to occur in the night side region
of the plasmapause and propagate along the magnetic field during the high years of
solar wind activity.

To identify the source of the Pc1 waves observed by CSES and SWARM, we use the
CCMC model [Pierrard et al., 2008] to obtain the variation of the position of the
plasmapause during this magnetic storm on August 26, 2018 (as shown in Figure 10).
Results show that the plasmapause moves outward at about UTC 23:00 on August 27,
and the *L* value reaches about 3 near local time 02:00. The red asterisk denotes the
position where Pc1 waves were observed by CSES. Moreover, based on the formula in
Carpenter and Anderson [1992] (shown as equation 1), the position of the




plasmapause is estimated at about *L*=2.98. Therefore, we suggest that the possible
sources of Pc1 waves are nearly located at the plasmapause, and this is consistent with
previous studies, that the outward expansion of the plasmasphere into the ring current
during the recovery phase of geomagnetic storms may generate EMIC waves, which
propagate along the background magnetic field to the ionosphere, and be observed
by multi-ground stations [Wentworth,1964; Cornwall et al., 1970; Russell & Thorne,
1970].
$$\hat{L}_{pp} = 5.6 - 0.46 \times max_{-24,-4}K_p \qquad (1)$$
According to the wave analysis performed using CSES and SWARM data, together with
ground station observations, we suggest that the satellites are close to the wave
injection regions in the Southern and northern hemisphere, during the recovery phase
of the storm. The incident waves propagate almost along the background magnetic
field, as transvers Alfven waves, which has long been predicted by theoretical studies,
although direct observations are rare. However, the ellipticity of the waves shows a
complex pattern, which is different from the polarizations of EMIC waves (LHP) in the
magnetosphere found by previous works [Fraser et al., 1975a, b; Erlandson et al.,
1990]. Theoretical studies predict that EMIC waves trigged near the Earth's magnetic
equator propagate toward the ionosphere, changing wave characteristics such as
ellipticity and wave normal angle when they pass through multicomponent plasma
[Denton, 2018; Johnson & Cheng, 1999; Kim & Johnson, 2016]. The mixed polarization
pattern observed in our case might either result from incident waves with complex
polarization pattern, or be attributed to the interference between the incident wave
and ducting waves in the ionospheric waveguide.
Joint magnetic field and electric field observations onboard CSES provide
unambiguous evidence that Pc1 waves propagate downward into the ionosphere in
the nearly conjugate ionospheric regions. Although the observations at north and
south are temporally separated by about 30 mins, it seems reasonable to infer that
the EMIC waves propagate northward and southward from the magnetic equatorial
region simultaneously, and wave reflection from the ionosphere is insignificant. Our
result is in accord with the CRRES satellite measurements reported by Loto'aniu et al.
(2005), which observed that outside a region of about +/-11° MLAT around the equator,
the Poynting vectors of the EMIC waves are directed away from the equator along the
magnetic field lines.
Pc1 waves sometimes have repetitive wave packet structures, which have been
explained by a bouncing wave packet model [e.g., Jacobs and Watanabe, 1964].
According to this model, a wave packet triggered in the equatorial region travels along



the magnetic field line, and is reflected between conjugate hemispheres. The Poynting
vector is an important parameter for establishing the propagation direction of wave
packet energy. CSES observations of Poynting vector in the ionospheric do not seem
to support this model.

## Conclusion

In this paper, using the simultaneous observations from CSES and SWARM satellites
and the ground geomagnetic stations data, we investigated the typical Pc1 waves in
the Northern and Southern ionospheric hemispheres. Our principal results are as
follows.
1. During the storm recovery phase on Aug. 27, 2018, the typical Pc1 waves were
recorded by the SGO stations on the ground for several hours. Meanwhile, the Pc1
waves were detected by the China Seismo-Electromagnetic Satellite (CSES) and
SWARM both in Northern and Southern hemispheres in the high latitude post-
midnight ionosphere region with a central frequency about 2 Hz.

2. In the field-aligned coordinate system, the power spectrum, ellipticity and normal
wave angle, Poynting vector are analyzed. Results show that the satellites observed
transverse Alfven waves with mixed polarizations, propagating almost parallel to the
background magnetic field downward, which imply the satellites were close to the
wave injection region in the ionosphere at about $L$=3. Attenuation of Pc1 wave power
at ground stations with increasing distance from $L$=3 also supports the idea that CSES
observes the wave activities near the injection region.

3. Furthermore, it is also found that the position of the plasmapause calculated by
CCMC model and equation of Carpenter and Anderson is almost at $L$=3. Therefore, we
suggest the possible sources of waves are near the plasmapause, which is consistent
with previous studies that the outward expansion of the plasmasphere into the ring
current during the recovery phase of geomagnetic storms may generate
electromagnetic ion cyclotron (EMIC) waves. Downward pointing Poynting fluxes
measured by CSES at nearly conjugate hemispheres suggest EMIC waves propagate
northward and southward simultaneously to the ionosphere at about $L$=3.

## Acknowledgments

The work is supported by NSFC grant 41904147, National Key Research and
Development Programs of Ministry of Science and Technology of the People´s Republic
of China (MOST) (2016YBF0501503, 2018YFC1503501). This research made use of the





data from CSES mission, a project funded by China National Space Administration (CNSA) and China Earthquake Administration (CEA). Additionally, thanks to ESA SWARM teams for providing SWARM FGM data, and SGO Webs for providing the SGO magnetic field data, and NASA CDA Web for providing the OMNI solar wind and magnetic field data, and NASA CMCC Web for providing the plasmapause simulation data.

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

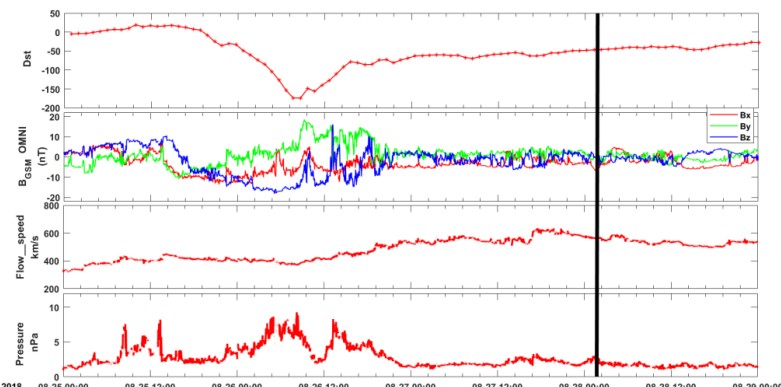


Figure 1. The solar wind conditions and geomagnetic index from Aug. 25 to 29, 2018.
From top to bottom: Dst index, interplanetary magnetic field, solar wind speed and
solar wind dynamic pressure, respectively. The occurrence of Pc1 waves is marked by
the black line.

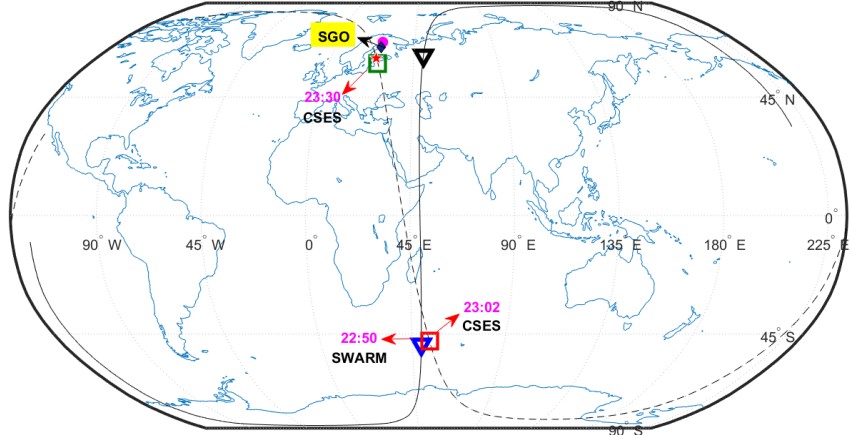






Figure 2. The locations of Pc1 waves observed by CSES (squares) and SWARM (triangles)
satellites. The pentagram, rhombus and circle represent three the SGO stations:
Nurmijärvi (NUR; *L* = 3.4, 57.1°N, 101.2°E, QD), Oulu (OUL; *L* = 4.5, 61.9°N, 104.1°E,
QD), and Sodankylä (SOD; *L* = 5.3, 64.3°N, 105.6°E, QD), respectively. The black dotted
and solid lines denote the trajectories of CSES and SWARMA satellites, respectively
and the red arrows represents three Pc1 wave observations.

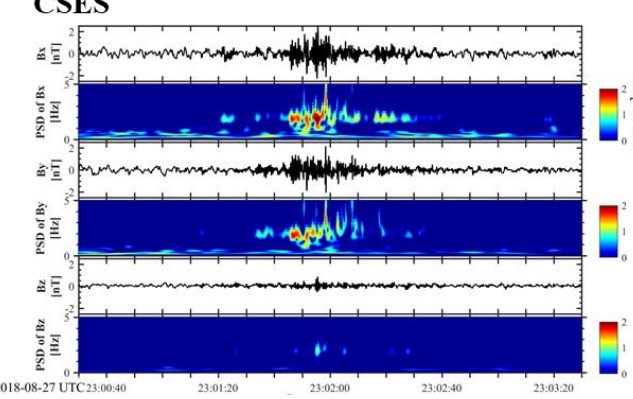


Figure 3. The power spectral densities (PSDs) of the magnetic fields during the Pc1
wave period (UTC 22:50-22:51) observed by SWARM-A.

Figure 4. The power spectral densities (PSDs) of the magnetic fields during the Pc1
wave period (UTC 23:01-23:02) observed by CSES.

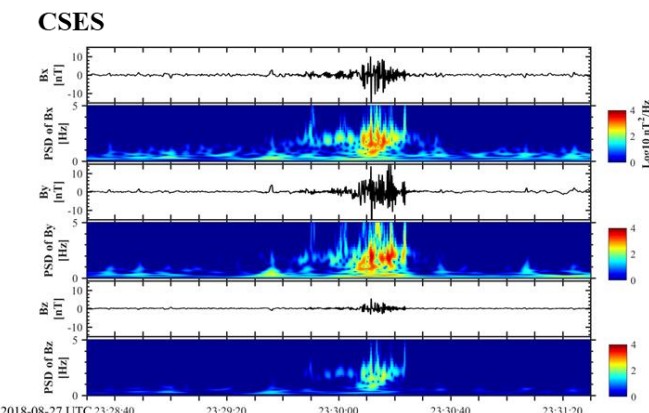

Figure 5. The power spectral densities (PSDs) of the magnetic fields during the Pc1 wave period (UTC 23:30-23:31) observed by CSES.

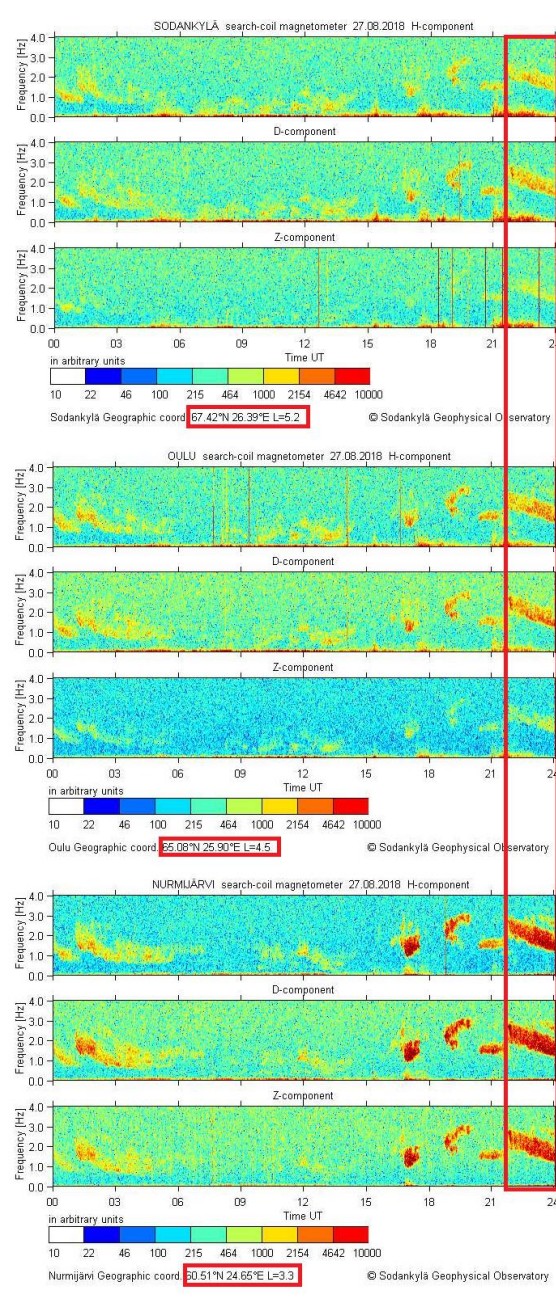

Figure 6. The power spectral densities (PSDs) of the magnetic fields during the Pc1
wave period (UTC 22:00-24:00) observed by SGO ground stations at different *L* shell
values.

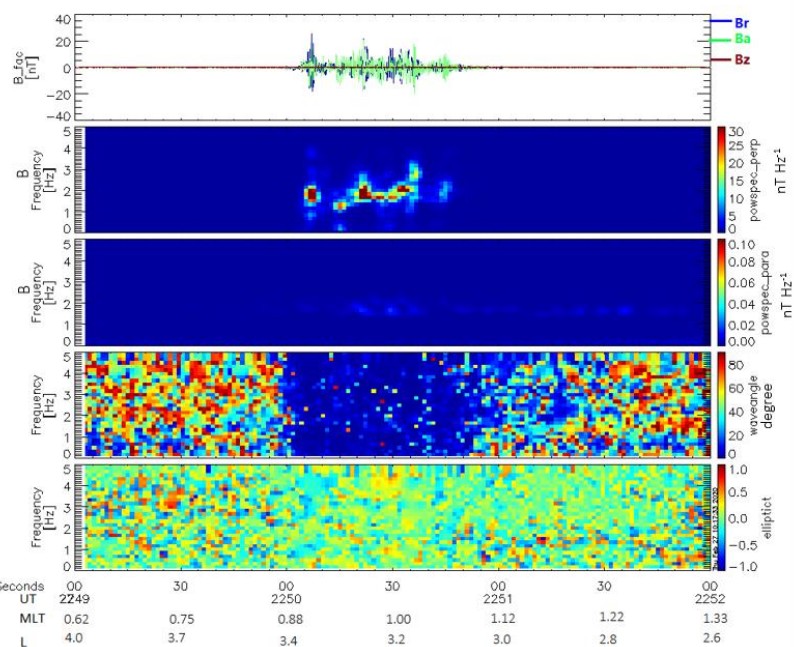

Figure 7. The wave propagation and polarization features of the Pc1 waves observed by SWARM. From top to bottom, magnetic field components (including perpendicular components Ba and Br marked in blue and green, parallel component Bz marked in red), wave power spectrum in perpendicular and parallel directions, wave normal angle and ellipticity computed by wave vector analysis of Means [1972]. (positive indicates right-handed polarization and negative indicates left-handed polarization).


Figure 8. The wave propagation and polarization features of the Pc1 waves observed



by CSES in the Southern hemisphere. From top to bottom, electric field components
(including perpendicular components Ea and Er marked in blue and green, parallel
component Ez marked in rad), electric wave power spectrum in perpendicular and
parallel directions; magnetic field components (including perpendicular components
Ba and Br marked in blue and green, parallel component Bz marked in rad), wave
power spectrum in perpendicular and parallel directions, magnetic wave normal angle
and ellipticity, the field-aligned Poynting fluxes.

Figure 9. The wave propagation and polarization features of the Pc1 waves observed by CSES in the Nothern hemisphere, same format as Figure 8.





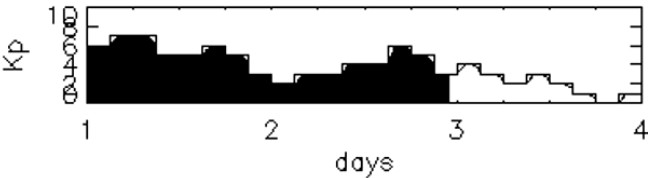

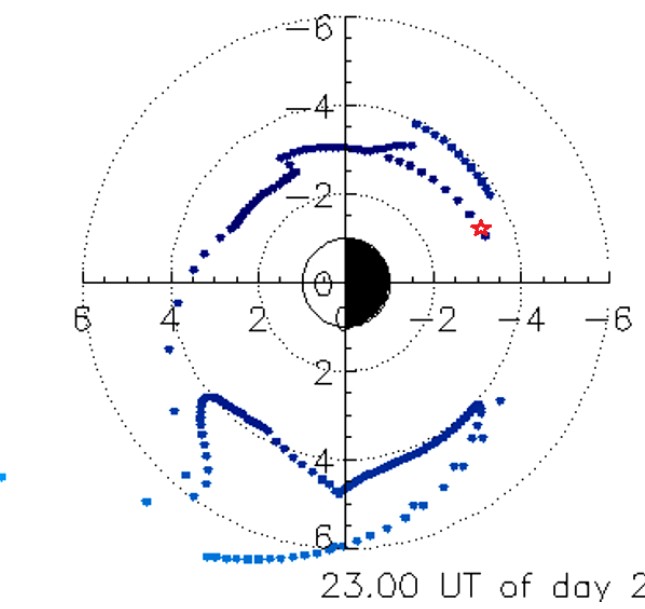

Figure 10. The Kp index (upper) and the simulated plasmapause location (lower) marked by blue dots at UTC 23:00 on August 27, 2018 from CCMC Web. The asterisk represents the conjugate location of Pc1 waves observed by CSES in the Southern hemisphere.