# Peer review of "Ionospheric Pc1 waves during a storm recovery phase"

_Annales Geophysicae, 2020_

## Referee Comment (RC1) · Anonymous Referee #1 · 24 Mar 2020

Review of "Ionospheric Pc1 waves during a storm recovery phase observed by CSES: by X. Gou et al.

I. General Comments

This paper reports observations of an interval of electromagnetic ion cyclotron (EMIC) waves in the Pc1 band that was observed in conjugate hemispheres by low-Earth orbiting (LEO) satellites and on the ground in the northern hemisphere. These observations, at both ends of a flux tube near L $\sim$ 3 within $\sim$ 30 minutes of each other during what appears to be a two-hour long interval of waves in both hemispheres, confirm many suggested features expected for EMIC waves generated near the magnetic equator.

[Figure]

This paper reviews several earlier studies of Pc1 waves, presents data from both the China Seismo-Electromagnetic Satellite (CSES) and SWARM-A as well as data from three ground-based pulsation magnetometer stations in Finland, and shows the results of one model of the location of the plasmapause at the time the waves were observed.

The presentation is mostly up to international standards and mostly clear, but many inaccuracies or errors are noted below, as are instances where technical language and English usage need improvement. All but one figure is sized adequately.

This manuscript may have potential value to the space science community after numerous corrections are made and possibly some additional work is done

II. Specific Comments

Lines 27-29: The authors might consider using a comparison of satellite and ground wave amplitude data to estimate the distance from the footpoint of the flux tube in which the waves originated to the location of the nearest ground station. If this can be done, it would be an excellent use of their data.

Lines 43-44: The citation of Wentworth (1964) here should be removed. The Wentworth paper makes no mention of hot ion temperature anisotropy, plasmapause, or ring current.

Lines 71-74. Satellites at LEO are well known to be unable to clearly detect Pc1 waves in the auroral zone. This is clearly expressed in section 2.2 of the Park et al. (2013) paper that the authors cite, so the authors must correct their statement. This limitation applies to the SWARM data as well (the algorithm used to identify events by Kim et al. (2018) also excludes data from the auroral zone), so the authors must also correct this statement. Many studies of Pc1 waves using high altitude satellite data have shown an increase in occurrence probability of Pc1 waves with increasing L out to almost the magnetopause.

Lines 106-107: A plot of the OMNI IMF data on August 27 and 28 using CDAWEB
shows that the IMF Bz component was NOT northward during this interval. During these 2 days it oscillated irregularly between positive and negative values. The text on line 107 thus needs to be corrected. The panel in Figure 1 showing IMF data does not show the Bz data clearly because of the compressed vertical scale. It only shows that the IMF Bz magnitude is smaller than it was during the main phase of the storm.

Line 109 and Figure 1. The black vertical line in Figure 1 is positioned at the wrong UT time, so Figure 1 needs to be corrected.

Line 138: change "~21:00" to "00:00" . Figure 6 shows 24 hours of data from each of the 3 stations listed.

Line 214: The Grison paper does not report "typical" magnetospheric EMIC waves. It presents 3 examples of EMIC waves that included triggered emissions, out of a total of only 4 such events observed during the entire Cluster mission from 2000 to 2008. The Pc1 waves reported in this manuscript have none of the characteristics of triggered emissions.

Line 220: The "CCMC model" needs to be better specified. Which model of the several models available at the CCMC was used in the study that is referenced here? Calling it a CCMC model is not appropriate.

Lines 219-224 and Figure 10: Figure 10 needs much more explanation. What distinguishes the blue and purple asterisks? What explains the rarity and large scatter of asterisks between 11 and 15 MLT? Are there 2 simultaneous plasmapauses between 02 and 05 MLT and between 16 and 21 MLT? How does this figure "show that the plasmapause moves outward at about UTC 23:00" ? Also, the "red asterisk" is actually a star, not an asterisk. The blue and purple symbols are asterisks.

Lines 302-306: Web addresses should be provided for each of the data sources listed here, and "CMCC" in line 305 should be changed to "CCMC"

References section: at least four reference citations are incomplete. They include, as

in line 322, the characters "n/a-n/a" but the actual page numbers are available on the appropriate JGR or Wiley web sites.

Lines 434-436: The pentagram, rhombus, and circle in Figure 1 are not visible unless this figure is greatly expanded. They instead appear as one oddly shaped dark blob. They may be visible if this figure is printed in a much larger format.

III. Corrections

Lines 17-18: Replace "satellites observed Pc1 waves exhibit" by "Pc1 waves observed by the satellites exhibited"

Line 49: Replace "magnetic activities" by "increased magnetic activity"

Line 72: Replace "dependence on the" by "dependent on"

Lines 75-76: Replace "preferable to occur during late recovery phase of the storm" by "and preferably occur during the late recovery phase of magnetic storms"

Line 123: Change "magnetometer HPM" to 'HPM Magnetometer"

Line 156: Change "LPH" to "LHP"

Line 168: Insert "and the" before "parallel component"

Line 173: Change "Figure 8-9" to "Figures 8 and 9"

Line 176: Change "the" to "that" before "wave normal angles"

Line 177: Replace "almost parallel propagate with" by "propagated almost parallel to"

Line 187: The words "On the other hand," do not seem to be appropriate here.

Line 197: Change "proves" to "confirms"

Line 286: Replace "activities" by "activity"

Lines 288-289: Change "by CCMC model and equation . . ." to "by the CCMC model

and the equation ...", and clearly specify the model.

---

## Referee Comment (RC2) · Anonymous Referee #2 · 4 Apr 2020

This paper presents new results from the CSES satellite of Pc1 observations in two hemispheres during the late stage of a geomagnetic storm. Magnetic and electric field data are presented, allowing a complete characterization of the waves in space. Comparison is made to magnetic results from the Swarm-A satellite and from Scandinavian ground stations near the satellite path. The source region of the waves, inferred to be EMIC waves, is inferred to be near the magnetic equator on a low L shell (near the plasmapause), with propagation from there toward both ground hemispheres determined from the Poynting vector. Ducting into the ionosphere is also discussed based on the ground data.

My major overall concern with the manuscript is that data from external to the mission is used in a manner that should have resulted in co-authorships for other teams. The

[Figure]

Swarm and ground data are used in a substantial way, so this would have seemed appropriate. Figure 6 indeed has copyright marks on it, suggesting it was taken from a website and that the data owners did not intend publication in this way. As such, I also make a comment that insufficient information is given about these data sources and their treatment. This being said, I feel it is an excellent article and worthy of publication. I have only minor comments on small errors: the writing is generally good and within what I would expect the copy editor to help with. These minor comments are listed only at the page level.

p.2 MAGAT seems to mean Magsat (maybe also called MagSat) which is not an acronym and does not need all capital letters

p.3 Similarly I am pretty sure Swarm is not an acronym. More importantly, References should be given for the instrumentation, much as for CSES' instruments. Then such details as bandwidth of the instruments etc. that may be needed for more detailed comparison of data could be had. Data treatment should also be mentioned. This applies also for ground data. As mentioned above, I feel that data from these sources was used to an extent that co-authorship should have been offered.

p.4 The rather long sentence about location ending in "Figure 2" (line 120) maybe should refer to Figure 3 since at the end the data is discussed. Anyway the sentence is long and unclear.

p.5 LPH should be LHP but maybe it should be stated somewhere that this means "left hand polarized"

p.6 line 200 should not start with "And". Also please give a reference for the analysis techniques.

References: several are out of order, of which I noted Hayashi, Iyemori. The Russell and Thorne reference should not be capitalized and was in Cosmic Electrodynamics, 1, 67-89 (in fact the title began with "On the structure. . ."

Figure 2: I found the symbols used for ground stations hard to see, maybe simply color code them

Figure 6: as noted above this appears to show copyrighted data or maybe the presentation format is copyrighted, anyway there is a problem. Also the color scale should not be shown three times as it is common to all plots.

Figure 7. This comment applies to other plots as well. Larger indications of what each subpanel shows would be helpful. Please put label boxes on the plots, there is lots of space in most of them to do so without obscuring meaningful data. It is hard to crank around to look at a tiny label on a color bar to know what one is looking at.

Figure 10. I am not sure what I am looking at in the lower panel. Presumably it is a projection of the plasmapause at the given time. Not sure why there are two dots at many local times (assuming it is local time, it is not labelled).

---

## Author Comment (AC1) · 7 Apr 2020

We thank the Anonymous Referee #1 for the helpful comments and suggestions. Attached, you can find the revised manuscript and our answers to your comments. The comments by the reviewer are in italics, and our responses are in Times New Roman. Corresponding changes are highlighted in the revised manuscript.

Please also note the supplement to this comment:
https://www.ann-geophys-discuss.net/angeo-2020-10/angeo-2020-10-AC1-supplement.zip

---

## Referee Comment (RC3) · Anonymous Referee #1 · 24 Apr 2020

Review of the second revised version of "Ionospheric Pc1 waves during a storm recovery phase observed by CSES" by X. Gou et al.

The authors have responded helpfully to nearly all of the comments and suggestions made by this reviewer. However, their responses to two points are unsatisfactory. The second of these points was also a concern of the other reviewer.

The first point concerns references to papers by Park et al. [2013] and Kim et al. [2014].

Lines 72-78: "According to the statistical analysis of CHAMP satellite data during one solar cycle, Park et al. [2013] found that Pc1 waves are mostly linearly polarized, having a peak occurrence at sub-auroral latitudes, and weakly dependent on magnetic

activity and the solar wind velocity. The Swarm data show a peak occurrence rate of Pc1 waves 75 at middle latitude including sub-auroral region. Moreover, these waves are linear 76 polarization dominated, propagating oblique to the background magnetic field, and 77 preferably occur during the late recovery phase of magnetic storms [Kim et al. 2018a]. "

The authors' reply is the following: "Surely, Park et al., 2013 and Kim et al., 2018 all proved that the peak occurrence rate of Pc1 waves is at midlatitude including sub-auroral region, so we used "sub-auroral region" instead of "auroral zone" (see line: 73)."

The authors' reply is technically correct, but only to the extent that data in the auroral zone was excluded from these two studies. This was stated explicitly in the Park et al. (2013) paper, and was implicit in the Kim et al. (2018) paper as well. If the authors' reply is taken literally, it contradicts numerous studies of Pc1 waves using high altitude spacecraft, which travel in regions where these waves are generated. The exclusion of data from the auroral zone in these two studies needs to be stated in this paper as well,

The second point concerns Figure 10 and its description in the text and figure caption.

Lines 225-233: "The blue dots correspond to the position of the plasmapause and the red star represents the conjugate location of Pc1 waves observed by CSES in the Southern hemisphere. From 11 to 21 MLT there is a plume rotating with the plasmasphere in the eastward direction. Such plumes are mostly formed during geomagnetic storm recovery phase [Pierrard and Cabrera, 2005]. Additionally, the plasma refilling process after the geomagnetic storms and substorms is included in this kinetic plasmasphere model. Between 02 and 05 MLT, two blue dots correspond to the inner edge of the refilling region and the outer edge of the plasmasphere and plasma refilling is expected in this intermediate region [Pierrard and Cabrera, 2005]."

First, the dots (asterisks?) appear to have two different colors: blue and purple. The

blue dots appear to trace the plume between 16 and 21 MLT, and two dots beyond L = 6 are blue, but all of the other dots appear to be purple.

Second, between 02 and 05 MLT there are two series of purple dots (not two dots).

This reviewer recommends changing one of the two colors of the dots/asterisks, and correcting the text to better match what is in this figure. As the other reviewer noted, local time should also be labeled (6, 12, 18, 0 or 24).
* * *

---

## Author Comment (AC2) · 24 Apr 2020

We thank the Anonymous Referee #2 for the valuable comments and suggestions. Attached, you can find the revised manuscript and our responses to your comments. The comments by the reviewer are in italics, and our responses are in Times New Roman. Corresponding changes are highlighted in the revised manuscript.

Please also note the supplement to this comment:
https://www.ann-geophys-discuss.net/angeo-2020-10/angeo-2020-10-AC2-supplement.zip

---

## Author Comment (AC3) · 6 May 2020

We would like to thank Referee #1 for the comments and suggestions. The files with our answers and the revised manuscript were already uploaded. The comments by the reviewer are in italics, and our responses are in Times New Roman. Corresponding changes are highlighted in the revised manuscript.

Please also note the supplement to this comment:
https://www.ann-geophys-discuss.net/angeo-2020-10/angeo-2020-10-AC3-supplement.zip

---

## Author Response (AR1)

Dear editor,

Thanks very much for your help. We would like to add the collaborator " Yiteng Zhang" as the co-correspondents, if possible. Looking forward to the publication of manuscript.

Sincerely,
Xiaochen Gou